# Spatially Explicit Reconstruction of Anthropogenic Grassland Cover Change in China from 1700 to 2000

**Fan Yang [1,2]** , **Fanneng He [1,\*]** and **Shicheng Li [3]**

1   Key Laboratory of Land Surface Pattern and Simulation, Institute of Geographic Sciences and Natural Resources Research, Chinese Academy of Sciences, Beijing 100101, China; yangf.17b@igsnrr.ac.cn
2   University of Chinese Academy of Sciences, Beijing 100049, China
3   School of Public Administration, China University of Geosciences, Wuhan 430074, China; lisc@cug.edu.cn
\*   Correspondence: hefn@igsnrr.ac.cn; Tel.: +86-10-64889066

**Abstract:** Long-term anthropogenic land use and land cover changes (LULCCs) are regarded as an important component of past global change. The past 300 years have witnessed dramatic changes in LULCC in China, and this has resulted in the large-scale conversion of natural vegetation to agricultural landscapes. Studies of past LULCC in China have mainly focused on cropland and forest; however, estimates of grassland cover remain rare due to the scarcity of grassland-related historical documents. Based on a qualitative analysis of trends in grassland cover in China and their driving forces, we devised different reconstruction methods for grassland cover in eastern and western China and then developed a 10 km database of grassland cover in China for the past 300 years. The grassland area in western China decreased from $295.54 \times 10^6$ ha in 1700 to $269.78 \times 10^6$ ha in 2000 due to the increase in population and cropland, especially in northeastern China (Heilongjiang, Jilin, and Liaoning), Gan-Ning, and Xinjiang. In eastern China, grassland is degraded secondary vegetation characterized by shrub grassland and meadow grassland, which is scattered in the hills and mountains; its area increased from $7.30 \times 10^6$ ha in 1700 to $16.43 \times 10^6$ ha in 1950 due to the increase in the degraded land caused by deforestation.

**Keywords:** land use and land cover change; grassland; subregional reconstruction; land reclamation; China

## 1. Introduction

Over the past, land use and land cover change (LULCC) has substantially altered the landscape of the Earth's terrestrial surface and continuously influenced climate change through a series of biogeophysical and biogeochemical mechanisms [1–7]. Long-term LULCC, therefore, has important implications for global environmental change. Substantial progress toward reconstructing historical LULCCs at different spatiotemporal scales has been made in recent decades.

There are four well-known global historical land-use scenarios: the History Database of the Global Environment (HYDE) [8]; the anthropogenic land cover change scenario from 8000 years ago to AD 1850 (KK10) [9]; the dataset of cropland and pasture for 1700–2007 (SAGE) [10,11]; and the global agricultural areas and land cover dataset for last millennium (PJ) [12]. Additionally, many regional historical LULCC reconstructions have been developed, including the anthropogenic deforestation dataset for 1000 BC–AD 1850 in Europe [13], land use for 1000 BC–AD 1500 in sub-Saharan Africa [14], cropland expansion and abandonment analysis for 1850–2016 in the continental United States [15], and land use for 1880–2010 in India [16].

Global historical land-use scenarios are dedicated to the reconstruction of long-term LULCC and provide estimates of LULCC at the global and continental scales. However, due to the use of different data

sources and methods of reconstruction, the results of these scenarios vary widely [8,9,17], and significant differences have been observed between the global scenarios and regional reconstructions [18–21]. This has resulted in a poor understanding of past global LULCC and demonstrates the need for promoted historical LULCC reconstructions at the regional level [22].

In China, the reconstruction of past LULCC has mainly focused on cropland [23–25] and forest [26,27]. However, most current studies on the history of grassland cover changes in China are qualitative analyses; quantitative reconstructions are rare because of the scarcity of grassland-related historical documents. Statistics on the area of grassland first became available in the middle of the 20th century, and satellite-based grassland monitoring began in the 1970s. Existing long-term grassland reconstructions are limited to a few time periods [28] or sporadic regions [29,30]. Many qualitative studies suggest that historically, large areas of grassland have been cleared as a result of expanded land reclamation in northern China [31–33]; however, the temporal and spatial changes of grassland in China remains unclear.

As one of the most widespread forms of land cover on Earth, grassland covers an area of $34.00 \times 10^8$ ha, accounting for 24% of all land area worldwide [34]. In China, grasslands are the largest terrestrial ecosystem, and the area of all types of natural grassland accounts for approximately 41% of the national land area [35]. Grassland resources also provide significant economic value and ecological function. Grasslands provide forage for animal production and play an important role in wind-breaking and sand fixation, as well as in water, soil, and biodiversity conservation since Chinese grasslands are mainly distributed in areas characterized by fragile ecological environments and/or significant topographic fluctuations [36]. Changes in grassland cover have caused a series of climatic and ecological environmental problems [37]. Thus, reconstructing long-term grassland cover in China is valuable for understanding past global environmental changes.

The past 300 years have witnessed dramatic LULCCs in China accompanied by rapid population growth and the extensive popularization of drought-tolerant crops from abroad, including corn and potatoes. During this time period, land reclamation has accelerated in the northern borderlands as well as in the hilly and mountainous areas in the south. This has resulted in the large-scale conversion of natural vegetation to agricultural landscapes and the generation of large areas of degraded land. Through decades of research, we have a good understanding of the changes in croplands and forests in China during this period [23,26]. In contrast, we know little about the spatiotemporal changes in grasslands over the past 300 years. Based on a qualitative analysis of the history of grassland changes in China, this study reconstructed historical grassland cover change by synthesizing the potential natural vegetation and data on croplands and forests over the past 300 years. Then, maps of grassland cover for the time period of 1700–2000 were created with a 10 km resolution. The results provide reliable regional data for modeling climate change and improving the global historical LULCC scenarios.

## 2. Materials and Methods

### 2.1. Data Sources

China has a long history and abundant historical documents. These historical materials contain considerable LULCC information and play an important role in understanding past LULCC. However, information that can be used to directly estimate the amount of grassland in China is rare. Therefore, in this study, existing historical data on croplands and forests were used to reconstruct historical grassland cover. The datasets used for the reconstruction are introduced as follows: Using Chinese historical tax-cropland area records for 1661–1893 and government inventories for 1913–1996, Ge et al. [38] estimated the provincial cropland area of China over the past 300 years by solving the problem of cropland conversion caused by taxation and the deliberate or inadvertent omission of cropland area during land surveys. Li et al. [23] transformed the provincial cropland area into a spatially explicit dataset—the Chinese historical cropland dataset (CHCD)—based on the suitability of land for cultivation obtained by quantifying factors closely related to the spatial distribution of

cropland (including surface slope, altitude, and climatic potential productivity). Based on historical documents (including official history, local chronicles, encyclopedias, travel notes, literator notes, official documents, and collected literati works), contemporary surveys, statistics, and previous studies, He et al. [39] first estimated the Chinese provincial forest area for 1700–2000 through the use of time section correction and multi-source data calibration. Subsequently, the authors devised a forested area allocation model—the Chinese historical forest dataset (CHFD)—and allocated the provincial forested area into grids at a resolution of 10 km based on the principle that the lands with high suitability for cultivation will be deforested first, with deforestation spreading to marginal lands with lower suitability for cultivation [26]. In summary, the CHCD and CHFD depend on trustworthy historical documents and appropriate reconstruction methods; and compared to historical documents and remote sensing-based data, these two datasets can capture historical changes in forest and cropland areas in China well at the provincial and grid scales. See Table 1 for more details.

**Table 1.** Details of historical cropland (CHCD) and forest (CHFD) data for China, remotely sensed data (China's land use/cover datasets, CLUDs), and global potential natural vegetation (GPNV) data.

| Data | Data Sources | Temporal Coverage | Spatial Resolution | Reference |
|------|-------------|-------------------|--------------------|-----------|
| CHCD | Historical documents | 1661–1996 | 10 km × 10 km | Ge et al. [38]; Li et al. [23] |
| CHFD | Historical documents | 1700–2000 | 10 km × 10 km | He et al. [39]; He et al. [26] |
| CLUDs | Satellite images | 1980–2015 | 1 km × 1 km | Liu et al. [40]; Ning et al. [41] |
| GPNV | Satellite images, BIOME3 | | 5′ × 5′ | Ramankutty and Foley [11] |

Remote sensing-derived LULCC data and potential natural vegetation data are important auxiliary data for reconstructing historical grassland cover (Table 1). Using the human–computer interactive interpretation method along with Landsat TM (Thematic Mapper)/ETM (Enhanced Thematic Mapper) digital images covering China, China's land use/cover datasets (CLUDs) were created [41,42]. The CLUDs contain six land use/cover types (accuracy greater than 94.3%)—cropland, forestland, grassland, waterbody, unused land, and built-up land—along with 25 subclasses (accuracy greater than 91.2%). These CLUDs are available on the Resource and Environment Data Cloud Platform (http://www.resdc.cn).

Potential natural vegetation refers to the vegetation that would most likely exist without anthropogenic activities. At present, the global potential natural vegetation (GPNV) map is widely used to explore historical LULCC [11–13,26]. The GPNV data were synthesized from the DISCover land cover dataset and the modeled natural vegetation [11]. The GPNV comprises 15 vegetation types at a 5 min resolution. Reliability evaluation showed that the GPNV spatial pattern for northeastern China is similar to the macro-scale native vegetation pattern [42]. Therefore, we believe that the GPNV map is applicable to China and used the GPNV data to replace the regions dominated by land use.

Topographic and meteorological data were also used in this study. The altitude data were obtained from the digital elevation model (DEM) of the Geospatial Data Cloud (http://www.gscloud.cn). On this basis, the slope data were calculated using ArcGIS 10.2 software. The meteorological data included the gridded data of annual precipitation and annual average temperature from 1980 to 2015 in China. Based on the daily observation data from more than 2400 meteorological stations in China, these data were generated by calculation and spatial interpolation (http://www.resdc.cn).

## 2.2. Regionalization of Grassland Cover Change in China

Large regional differences in the spatial patterns and changes in grassland cover in China occur due to the limitations of the natural environment (e.g., topography and climate) and the impacts of anthropogenic activity. Statistics show that 79% of grasslands in China occur in the northern temperate zone and the Qinghai-Tibetan plateau alpine zone (I and II in Figure 1) [35]. Historically, these two regions have mainly been inhabited by nomads, and the nomadic way of production has not yet caused serious damage to grassland vegetation over a long period of time [32]. However, in the course of securing the borderlands in western China, successive dynastic governments vigorously developed

agriculture in the northern temperate zone, resulting in the extensive conversion of grasslands to croplands. Over the past 2000 years, this region has experienced four large-scale immigration and cultivation events, including during the Qin-Han, Sui-Tang, and the mid-Qing dynasties as well as throughout the mid-20th century [31,33]. Although the cold-dry climate of the Qinghai-Tibetan plateau has seriously hindered the development of agriculture in the past, grassland reclamation has still occurred in a few valleys, including the Yarlung Zangbo River and Huangshui River valleys [29,43]. As a result, the expansion of cropland is the main driver of decreases in the grassland area in the northern temperate zone and the Qinghai-Tibetan plateau alpine zone.

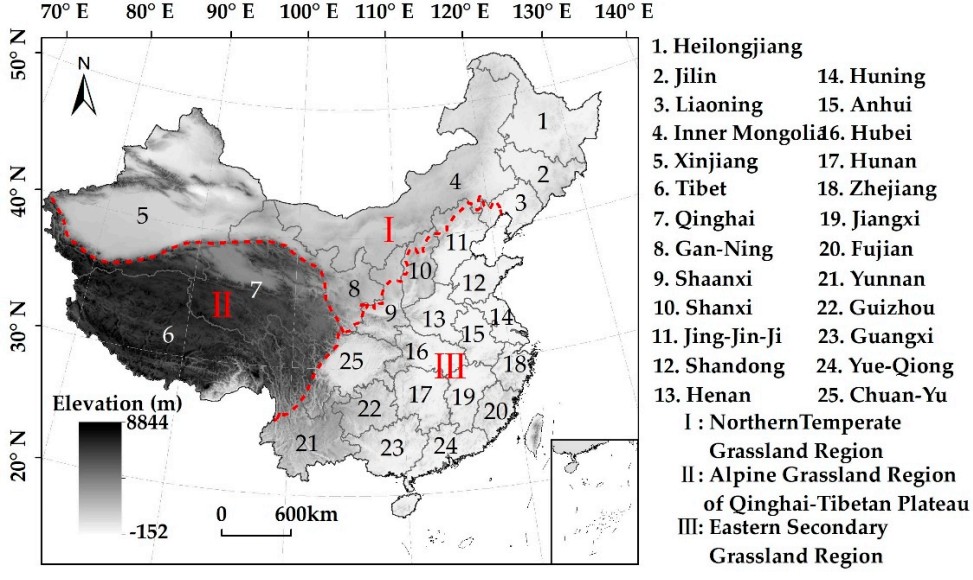

**Figure 1.** Location of the study area.

In eastern China, grassland vegetation is dominated by non-zonal secondary growth (III in Figure 1), which accounts for 21% of the total grassland area in China [35]. Compared to western China, grassland in eastern China is characterized by greater diversity and complexity in both distribution and succession [44]. Before agriculture emerged, the landscape in eastern China mainly comprised zonal forest vegetation with a few non-zonal secondary grasslands. Previous studies suggest that large-scale anthropogenic deforestation in the past generated a large area of "degraded land", apart from parts of the deforested land turning into cropland [45]; secondary grassland then emerged in part of the degraded land in the suitable natural habitat [44–46]. Therefore, it is clear that the area of secondary grassland increased as anthropogenic deforestation expanded.

Considering that trends in grassland cover are consistent between the northern temperate zone and the Qinghai-Tibetan plateau alpine zone, these two regions were merged into the western region. Reconstruction methods for historical grassland areas were then devised based on these two divisions (eastern and western China). To facilitate analyses of grassland cover change at the provincial scale, we used the 25 provincial units adopted by the CHCD and CHFD (Figure 1).

*2.3. Methodology*

The scheme for reconstructing grassland cover in China from 1700 to 2000 is illustrated in Figure 2. In western China, considering that changes in grassland cover have primarily been influenced by land reclamation in the past, we used the proportion of grid-based cropland cover out of the potential extent of natural grassland (PENG) to capture the change in historical grassland cover. For eastern China, changes in secondary grassland cover are primarily the result of interactions between anthropogenic deforestation activities and natural habitats. This study developed a grassland estimation scenario for this region based on these two factors, given that climate variability in China over the past 300 years has been relatively minor [47]. This scenario is based on two assumptions: (1) The habitat range of

current eastern secondary grassland among the potential extent of natural forest (PENF) is the historical possible distribution range of secondary grassland; and (2) the parts of non-forest and non-cropland in this habitat range for each period are regarded as secondary grassland.

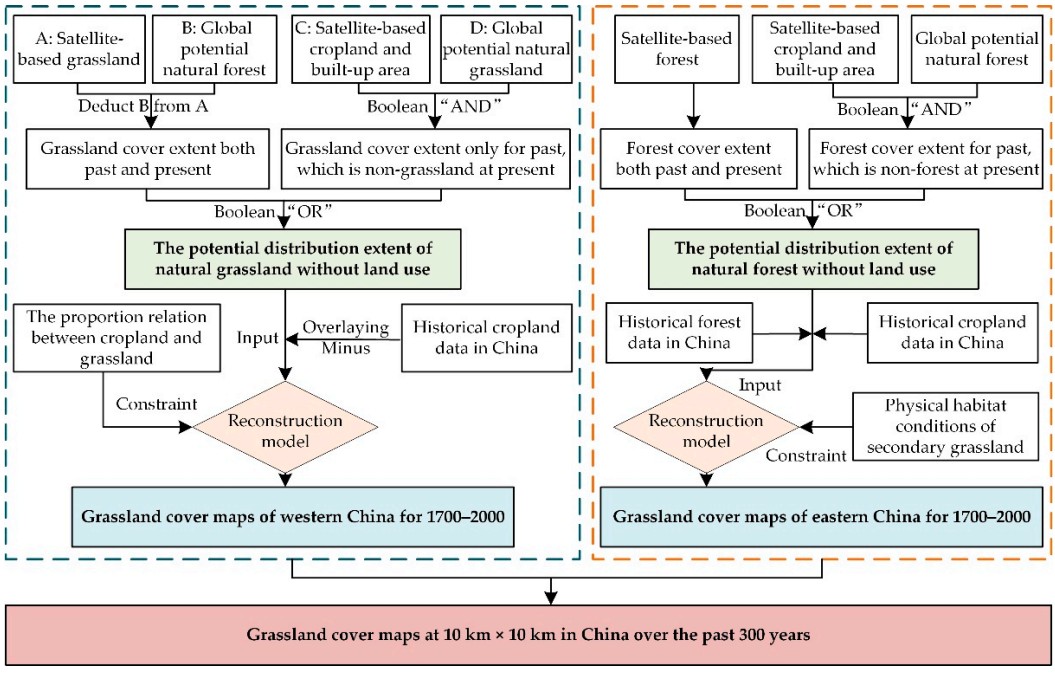

**Figure 2.** Scheme for reconstructing grassland cover in China from 1700 to 2000.

### 2.3.1. Determination of the Potential Extent of Natural Forest and Grassland without Anthropogenic Activities

Several studies have demonstrated that most contemporary grassland regions monitored by remote sensing in China are historically grassland regions, and small parts might also be forested [48]. Similarly, most contemporary forest regions detected using remote sensing were also forested regions in the past [26]. Non-forest and non-grassland regions detected by remote sensing, especially croplands and built-up areas, might have also been forests or grasslands in the past [42,49]. Based on this information, we developed a method using CLUDs and GPNV data to determine PENG and PENF in China.

The method used in this study to determine PENG and PENF in the absence of land reclamation involved the following steps: (1) We first obtained data for the grid of contemporary forest, grassland cropland, built-up areas, and potential grassland and forest cover. (2) After potential forest was subtracted from the grid data of contemporary grassland, the remainder was classified as both the past and present areas of grassland distribution. The spatial pattern of the contemporary forest was regarded as the distribution range of forest in both the past and present. (3) Potential forest and grassland cover were superimposed over the current non-forest and non-grassland cover, and the overlapped grid cells were classified as historical forest cover or historical grassland cover in the non-forest and non-grassland regions. (4) Finally, the forest and grassland cover obtained in steps (2) and (3) were combined to enable the determination of PENG and PENF without land reclamation.

### 2.3.2. Reconstruction Method of Grassland Cover across Western China

Grassland in western China was reconstructed using the method of Yang et al. [42]. In China, both in the past and at present, croplands are mainly distributed around urban built-up areas and rural settlements; the statistics of remote sensing-derived LULCC indicate that the proportion of cropland in western China was greater than or equal to 90% at a resolution of 10 km, the corresponding grassland coverage was mostly less than 1% because the remainder was generally occupied by residential

land [42]. Therefore, this study assumed historical grassland coverage to be zero in a grid if the proportion of cropland was greater than 90% in western China.

Based on this assumption, we first established PENG without land use and extracted historical cropland data across western China from CHCD. Grassland cover for 1700–2000 was further reconstructed by overlaying cropland cover over PENG. The equation used to determine the grassland area for western China in grid j (10 km resolution) in year t [$G_W(j, t)$] takes the form:

$$G_W(j,\ t) = \times \left[ \sum_{i=1}^{100} G_P(i,\ j,\ t) - C(j,\ t) \right] \tag{1}$$

where $G_P(i, j, t)$ denotes the $i^{th}$ PENG (1 km) in grid j in year t; $C(j, t)$ refers to the cropland area in grid j in year t; and $\alpha$ denotes the ratio of cropland to grassland. If $C(j, t)$ is greater than 90 km$^2$, the value of $\alpha$ is zero otherwise the value is one.

### 2.3.3. Reconstruction Method of Grassland Cover across Eastern China

In addition to anthropogenic deforestation, natural habitats have played important roles in changes in secondary grassland cover. Therefore, determining the common habitats suitable for the growth of secondary grassland is essential to reconstruct grassland cover in eastern China. Zhang et al. [50] explored the relationships between habitat and various grassland vegetation types by analyzing current remote sensing-derived grassland cover data along with corresponding topographic data (altitude and slope) and meteorological data (annual precipitation and annual average temperature). The results suggested that grassland vegetation in eastern China mainly comprises shrub grassland and meadow grassland, and their habitats are similar. These two types of grassland are located in the same topography; shrub grassland is mainly distributed in areas characterized by high temperature and precipitation, whereas meadow grassland is primarily distributed in areas with moderate temperature and precipitation. The details of these land cover types are given in Table 2.

**Table 2.** Habitat characteristics of shrub grassland and meadow grassland [50].

| Type | Temperature (°C) | Precipitation (mm) | Altitude (m) | Slope (°) |
|---|---|---|---|---|
| Shrub grassland | 8.4–13.2 (90%) | >600 (90%) | 300–2200 (85%) | <25 (90.22%) |
| Meadow grassland | 2–10 (90%) | >550 (90%) | 300–2200 (90%) | <25 (90.22%) |

Note: 90%, 85%, and 90.22% refer to the proportions of grassland areas in this grassland vegetation type within the corresponding temperature, precipitation, altitude, and slope range in the statistical analysis of contemporary data (grassland cover, topographic, and meteorological data).

The possible extent of secondary grassland vegetation in the PENF was determined based on the above analysis of the habitat of this land cover type. The forest and cropland data for different years across eastern China were then extracted from the CHFD and CHCD, respectively. Finally, we combined the possible extent of secondary grassland and the extracted historical forest and cropland data to reconstruct grassland cover for 1700–2000 in eastern China as follows:

$$G_E(j,\ t) = \left[ \sum_{i=1}^{100} SG_P(i,\ j,\ t) - C(j,\ t) - F(j,\ t) \right] \tag{2}$$

where $G_E(j, t)$ denotes the grassland area for eastern China in grid j (10 km) in year t; $SG_P(i, j, t)$ refers to the $i^{th}$ possible extent of secondary grassland (1 km) in grid j in year t; and $F(j, t)$ denotes the forested area in grid j in year t.

## 3. Results

### 3.1. Potential Extents of Natural Forest and Grassland

PENG and PENF are shown in Figure 3. In the absence of anthropogenic activities, PENG covered an area of $320.85 \times 10^6$ ha and was mainly distributed in western and northern China (Figure 3a).

Eastern China was dominated by forest cover without land reclamation, and the area of PENF accounted for $397.12 \times 10^6$ ha (Figure 3b).

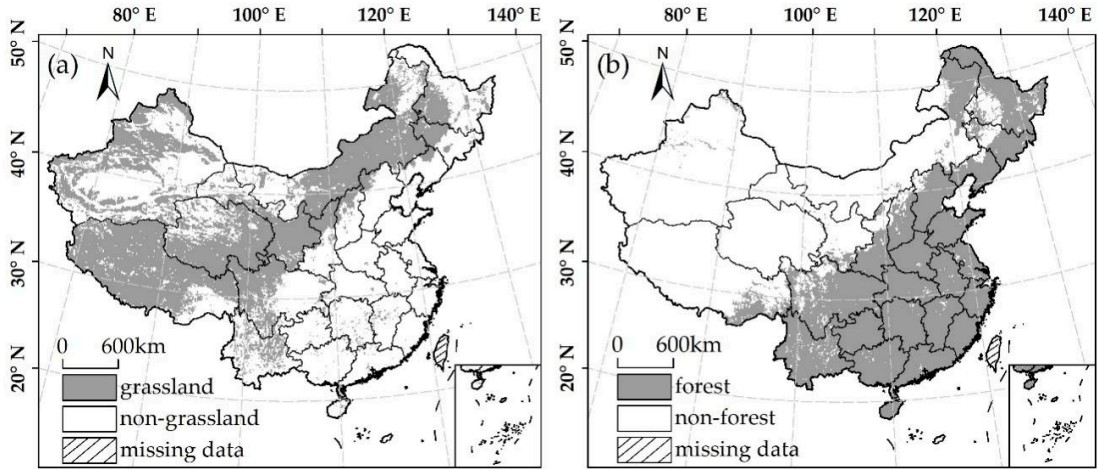

**Figure 3.** The potential extents of natural (**a**) grassland (PENG) and (**b**) forest (PENF) without land reclamation.

### 3.2. Total and Regional Grassland Area from 1700 to 2000

Figure 4 shows the total grassland area for China's mainland from 1700 to 2000. The area of grassland continuously decreased over time because of land reclamation activities. The reconstruction results suggest two phases of change in the grassland area: a slow decrease from 1700 to 1950 followed by a rapid decrease from 1950 to 2000. The grassland area decreased from $302.84 \times 10^6$ ha in 1700 to $296.74 \times 10^6$ ha in 1950, representing a total loss of $6.10 \times 10^6$ ha at an annual loss rate (ALR) of $2.44 \times 10^4$ ha. In 2000, the grassland area was $278.80 \times 10^6$ ha; this translates to a loss of $17.94 \times 10^6$ ha between 1950 and 2000 at an ALR of $35.87 \times 10^4$ ha.

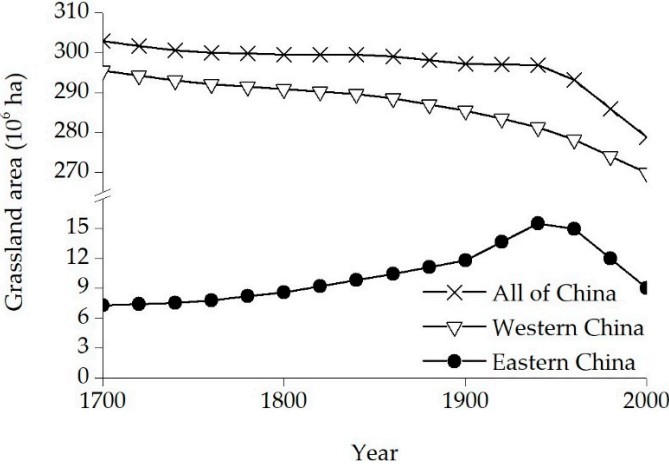

**Figure 4.** Changes in the total and regional grassland areas from 1700 to 2000.

At the regional scale, grassland areas in western and eastern China have shown contrasting trends over the last 300 years. Grassland cover in western China exhibited a continuous decrease in this time period with three distinct phases: a rapid decrease between 1700 and 1750, a slight decrease between 1750 and 1850, and a sharp decrease between 1850 and 2000 (Figure 4), with corresponding losses in the grassland at $3.07 \times 10^6$, $3.13 \times 10^6$, and $19.56 \times 10^6$ ha, respectively, and ALRs of $6.13 \times 10^4$, $3.13 \times 10^4$, and $13.04 \times 10^4$ ha, respectively. The grassland area in eastern China exhibited a two-stage trend (Figure 4), a gradual increase in the prophase and a rapid decrease in the anaphase. The area occupied by grasslands in this region increased from $7.30 \times 10^6$ ha in 1700 to $16.43 \times 10^6$ ha in 1950,

representing an increase of $9.12 \times 10^6$ ha over this 250-year period. In 2000, the grassland area was $9.03 \times 10^6$ ha; thus, $7.40 \times 10^6$ ha of grassland was lost between 1950 and 2000.

### 3.3. Spatial Distribution of Grassland Cover

Figure 5 illustrates the spatial pattern of grassland cover across China from 1700 to 2000. The spatial distribution of grassland over time generally remained constant; western China was dominated by grassland cover because of environmental conditions (e.g., climate, topography, and soil) and limited population levels. However, the reconstruction results indicated large-scale decreases in grassland areas in northeastern China (Heilongjiang, Jilin, and Liaoning), Xinjiang, and the Gan-Ning region. In eastern China, grassland was mainly scattered in the hills and mountains with a gradual increase in area over time.

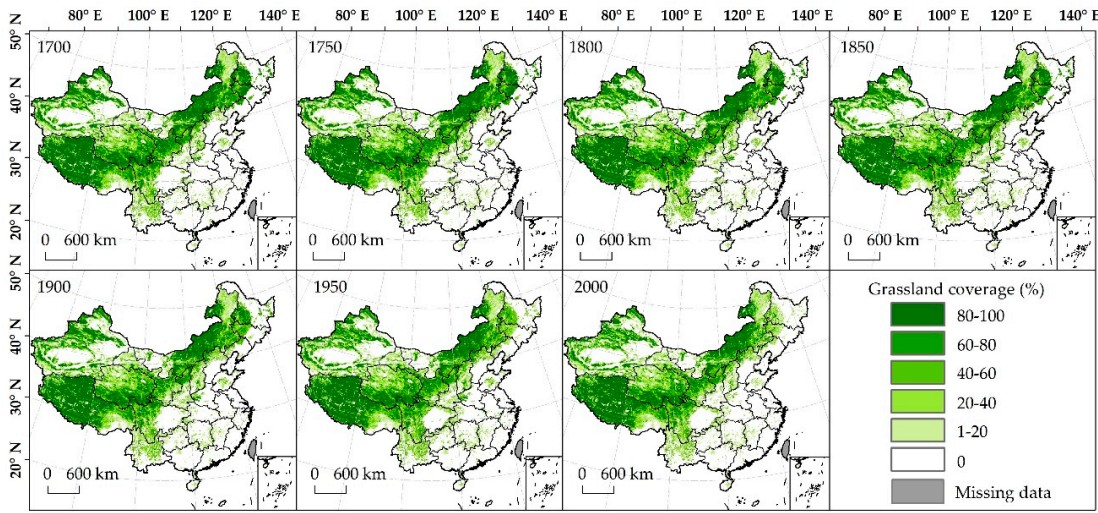

**Figure 5.** Spatial distribution of grassland cover in China from 1700 to 2000.

## 4. Discussion

### 4.1. Analysis of Grassland Cover Change Over the Past 300 Years

While it is difficult to quantitatively estimate the area of grassland in China based on historical documents, this study can use these materials to qualitatively assess our reconstruction. The grassland area across western China decreased slowly before the mid-19th century, and the clearing of grassland mainly occurred in Gan-Ning, Inner Mongolia, and northeastern China (Figure 6a). Specifically, the emperors Kangxi and Yongzheng (1661–1735) exempted all taxes except the basic ones to encourage farmers in the Gan-Ning region to open up grasslands [31,51]. This resulted in the large-scale conversion of grasslands into croplands in the central and eastern parts of the Gan-Ning region as enthusiasm for land reclamation was greatly enhanced. For instance, in some local areas, cropland area increased by three times compared to the original cropland area [52]. Throughout this period, refugees entered Inner Mongolia and northeastern China, despite the government's implementation of the "Prohibit reclamation in northeast China" policy [42]. As a result, the population of northeastern China increased from 1.01 million people in 1776 to 4.19 million people in 1850 [53]. Agriculture developed alongside this population growth, and the increase in cropland was mainly derived from grassland (Figure 6a).

Following the mid-19th century, the resultant limited cropland resources struggled to sustain the level of population, and the northern border areas were successively invaded by Russia and Japan. Against this background of internal revolt and foreign invasion, the Qing government began to encourage immigration into borderlands for the purposes of consolidating the borderlands and relieving conflicts between the rapidly increasing population and limited land resources [54]. Therefore, a large number of immigrants moved into this border area, resulting in the large-scale conversion of grasslands into croplands (Figure 6b) [31].

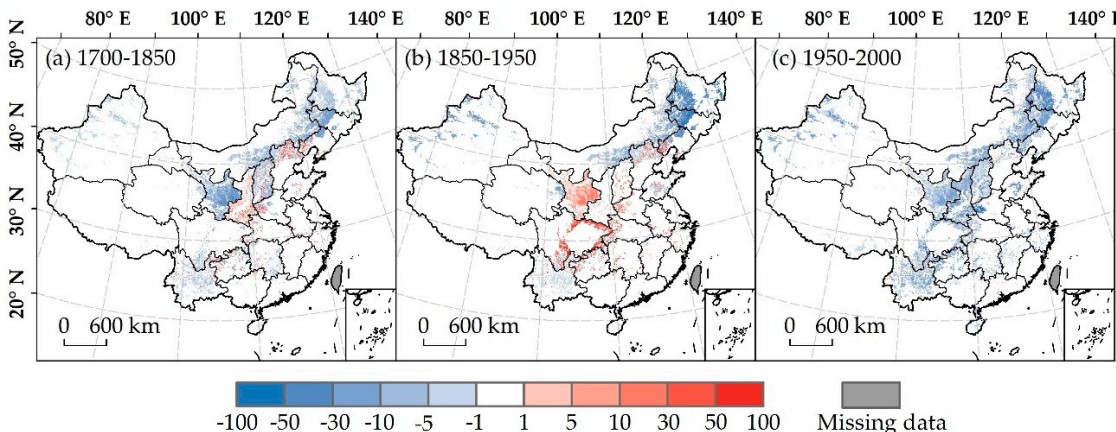

**Figure 6.** Net changes in grassland cover in China from 1700 to 2000.

Data suggest that the grassland area in western China has remained in a stage of continuous decrease since the mid-20th century (Figure 6c). Indeed, Fan et al. [55] suggested that 18.2% of current cropland in China was derived from grassland, and western China experienced three stages of large-scale grassland reclamation throughout this period: the 1950s, the mid-1960s, and the early 1970s.

Secondary grassland increased continuously from the Qing dynasty to the Republic of China (Figure 6a,b). Due to population growth, natural disasters, and chaos caused by wars, some people were driven from their homes, settled in simple tents in the mountains, and lived by land reclamation. These immigrants were thus called shed people. Historically, changes in grassland have been significantly influenced by the shed people. The population increased geometrically since the Qing dynasty. The population of China increased from 160 million in the late 17th century to 400 million by the middle of the 19th century [53]. A large number of people were thus forced into the mountains to make a living. At the same time, crops from overseas that were suitable for planting in the mountains including corn and potatoes became available, leading to intensified land reclamation in these regions [33,56]. The soil fertility in mountainous areas was low; as a result, slope land was frequently abandoned, and new cropland was established. This led to large-scale deforestation in the mountains and the emergence of further secondary grassland on parts of the deforested land. Our reconstructions show that the area of secondary grassland gradually increased in the mountainous areas, including the Yanshan, Funiushan, and Qinling mountains as well as the area surrounding the Sichuan basin, which has been deeply affected by continuous anthropogenic deforestation (Figure 6a,b).

Since the 1950s, sustained reforestation and afforestation efforts have resulted in an increase in forest cover. Large-scale afforestation was implemented by the Chinese government to meet the needs of the economic reconstruction from the 1950s to the 1980s [57]. After the 1980s, timber production was no longer the primary goal of forest management, and the government introduced a series of ecological programs aimed at protecting forest resources [58]. As a result, secondary grassland decreased gradually as forest cover grew (Figure 6c).

The results show that our reconstructed changes in grassland cover over the past 300 years are highly consistent with the information recorded in historical documents, including information on policies, population growth, immigration, and land reclamation.

### 4.2. Comparisons with CLUDs and the Vegetation Atlas of China

We attempted to verify our model results via comparison with remote sensing-based and field investigation-derived grassland data. Two available contemporary LULCC datasets in China contain data on grassland, CLUDs and the vegetation atlas of China (VAC). Based on Landsat TM/ETM digital images, the CLUDs were developed by using visual interpretation against field investigation data [40,41], and this dataset has strong timeliness. The VAC, which has a scale of 1:1 million, was published in 2001 [59]. This map was compiled by multiple scientific institutions based on a large

number of field investigation data throughout the country over the past 50 years and supplemented by remote sensing data. Compared to CLUDs, it has the advantages of high accuracy and less time taken.

Using the CLUDs and VAC, the reliability of the reconstructed grassland data was evaluated at the gross, provincial, and grid scales. The grassland area in the VAC, CLUDs, and our reconstruction in 2000 was $280.00 \times 10^6$ ha [36], $290 \times 10^6$ ha, and $278.80 \times 10^6$ ha, respectively. Thus, our reconstructed area of total grassland was closer to the VAC estimate than the CLUDs value. This is likely because the CHCD and CHFD were the main data sources used to reconstruct historical grassland cover in this study. The cropland and forest data for 1950–2000 in the CHCD and CHFD were derived from national statistics based on field investigations; thus, the data acquisition methods on which national statistics are based are more similar to those of the VAC than those of the CLUDs.

Figure 7 presents the relationships between our reconstruction and contemporary LULCC data at the provincial scale. The linear regression between the CLUDs and VAC grassland areas (Figure 7a) has an $r^2$ value of 0.990 and a slope of 1.034, indicating a strong correlation between them. Strong correlations were also observed between the provincial grassland areas of the CLUDs and this study ($r^2 = 0.998$; slope = 1.014; Figure 7b) and between those of the VAC and this study ($r^2 = 0.988$; slope = 0.961; Figure 7c). The correlations among these datasets with the provincial grassland area less than $10.00 \times 10^6$ ha are shown in Figure 7d–f. The values of $r^2$ between the provincial grassland areas of the CLUDs and VAC, the CLUDs and this study, and the VAC and this study were 0.767, 0.869, and 0.759, respectively, while the slopes were 0.790, 0.643, and 0.545, respectively. Overall, the correlations between them were fairly good, indicating that the reconstructed provincial grassland area in this study was close to those estimated in CLUDs and VAC.

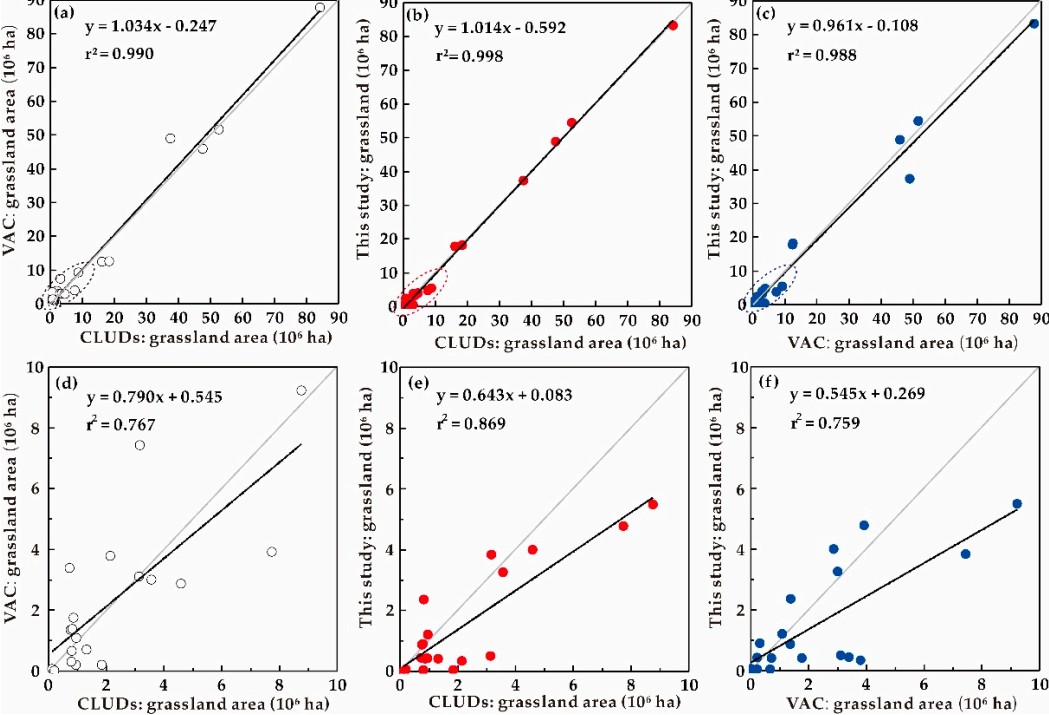

**Figure 7.** Provincial comparisons of grassland areas in 2000 derived from CLUDs (China's land use/cover datasets), our reconstruction, and VAC (Vegetation Atlas of China). The correlations at the provincial scale: (**a**) the correlation between the VAC and CLUDs; (**b**) the correlation between our reconstruction and CLUDs; (**c**) the correlation between our reconstruction and VAC. The correlations among these datasets with the provincial grassland area less than $10.00 \times 106$ ha: (**d**) the correlation between the VAC and CLUDs; (**e**) the correlation between our reconstruction and CLUDs; (**f**) the correlation between our reconstruction and VAC.

The CLUDs grassland distribution was compared to our reconstruction at the grid-scale because data in both are in raster format. Figure 8 shows the overall spatial patterns of grassland cover in these two datasets as well as the differences between them in 2000. The spatial distribution of grassland in our reconstruction generally agreed with the CLUDs distribution; both indicated that western China was dominated by grassland cover (Figure 8a,b). Positive differences (i.e., the reconstructed grassland area is greater than the CLUDs area) are distributed mainly in the western part of northeastern China, Yinshan Mountain in Inner Mongolia, northern Jing-Jin-Ji, Shanxi, Shaanxi, and southern Chuan-Yu (Figure 8c). Negative differences (i.e., the reconstructed grassland area is less than the CLUDs area) are distributed mainly in Taihang Mountain on the border of Shanxi and Hebei, southern Shaanxi, and western Guizhou.

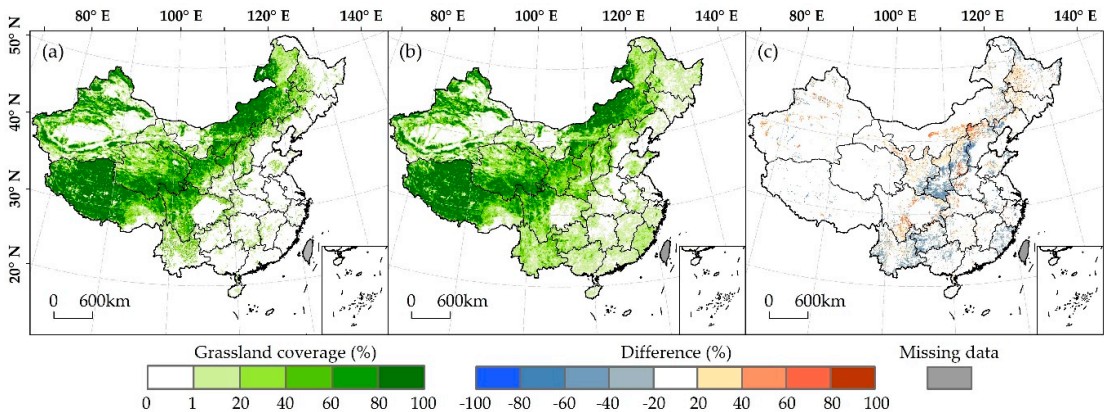

**Figure 8.** Spatial patterns of grassland cover in 2000: our reconstruction (**a**), CLUDs (**b**), and the difference between them (**c**).

The percentages of grid cells with various ranges of differences between the reconstructed and CLUDs grassland cover are shown in Table 3. Grid cells with differences of −10–0% accounted for 58.55% of all grid cells, while 17.76% of grid cells had differences of 0–10%. Grids with differences exceeding 50% accounted for only 1.12% of all grid cells, while those with differences lower than −50% accounted for only 1.20%. Thus, a larger absolute value of difference corresponded to a smaller percentage of grid cells (Table 3).

**Table 3.** Statistical classifications of the differences in grassland cover in 2000 between our reconstruction and CLUDs.

| Difference (%) | −100 to −50 | −50 to −40 | −40 to −30 | −30 to −20 | −20 to −10 | −10 to 0 |
|---|---|---|---|---|---|---|
| Percentage (%) | 1.20 | 1.15 | 1.82 | 3.23 | 6.09 | 58.55 |
| Difference (%) | 0–10 | 10–20 | 20–30 | 30–40 | 40–50 | 50–100 |
| Percentage (%) | 17.76 | 3.94 | 2.57 | 1.59 | 0.98 | 1.12 |

### 4.3. Prospects and Caveats of the Reconstruction Methods

Given the large differences in changes in grassland cover between western and eastern China, this study used zoning to reconstruct historical grassland cover. Based on comparison with historical documents and previous studies, the results of our reconstruction methods well capture the history of grassland changes in China. Due to the complexity of the changes in grassland cover in eastern China, we introduced the concept of habitat and utilized this concept to constrain the possible extent of secondary grassland in eastern China in the past. At present, this approach allows us to distinguish secondary grassland from the large amount of "degraded land" in eastern China.

Notably, China has a long history and a vast territory with disparate natural and human environments among regions, resulting in different land-use practices. The current zoning approach is not sufficient to accurately capture historical grassland cover; and the uncertainty inherent in data

sources (including the CHCD, CHFD, and GPNV) can also impact the results. Thus, it should be cautious about applying the dataset in this study at local research. The availability of even more information on changes in grassland cover at the regional scale would contribute to the division of more subregions in the future and the development of more reconstruction methods, which would more accurately reflect historic spatiotemporal changes in Chinese grasslands.

The method for reconstructing grassland cover in western China reflects the fact that changes in grassland area in this region have primarily been forced by land reclamation. However, changes in this land cover have also been influenced by numerous natural and cultural factors, including climate change, desertification, war, and the development of built-up areas. For instance, historical anthropogenic activities coupled with climate change accelerated desertification and greatly affected the grassland area [60]. It remains a challenge to provide quantitative information on these factors with respect to past land cover and land use.

For eastern China, we proposed a method for reconstructing grassland cover based on limited knowledge. As shown in Figure 7d–f, the uncertainties in the grassland area obtained by our method in eastern China are higher than those in western China. However, these uncertainties for eastern China have little effect on the total grassland area and its spatial pattern because grassland cover is mainly distributed in western China. Overall, the history of changes in secondary grassland in eastern China requires further exploration.

Based on some reasonable assumptions, this study determined PENG and PENF by utilizing GPNV and CLUDs. Numerous scholars have also presented quantitative reconstructions of past land cover based on archaeological and paleoecological records [61–64]. Although these records document the time history of anthropogenic land cover at numerous individual sites, it remains difficult to reconstruct long-term, spatially explicit maps of grassland cover across China at 10 km resolution using archaeological and paleoecological records. Therefore, the synthesis of multiple data sources, including archaeological data, pollen data, historical documents, statistics, and remote sensing images, shows great potential for reconstructing changes in historical land cover.

## 5. Conclusions

Historically, eastern and western China have exhibited large differences in changes in grassland cover. Based on these differences, this study devised methods for the reconstruction of historical grassland cover in eastern and western China and reconstructed grassland cover for 1700 to 2000. Over the past 300 years, the grassland area in China continuously decreased, and the total decrease in area was $24.03 \times 10^6$ ha. Population growth, agricultural expansion, and large scale deforestation were identified as causes of the grassland cover change. Meanwhile, traditional extensive land use over long time periods in China has resulted in a large number of degraded land while transforming natural vegetation into agricultural land. Through comparison with historical document-based qualitative studies, remote sensing-based CLUDs for 2000, and field investigation-derived VAC for 2001, we evaluated the reliability of our reconstructed grassland data at different spatial scales. The results suggest that our method for reconstructing historical grassland cover is reliable, and the reconstructed grassland cover maps accurately reflect the spatiotemporal patterns of Chinese grasslands in the past.

**Author Contributions:** Conceptualization, F.Y., F.H., and S.L.; methodology, F.Y.; formal analysis, F.Y.; data curation, S.L.; writing—original draft preparation, F.Y.; writing—review and editing, F.Y., F.H., and S.L.; funding acquisition, F.H. All authors have read and agreed to the published version of the manuscript.

**Funding:** This research was funded by the National Key Research and Development Program of China (Grant No. 2017YFA0603304), the National Natural Science Foundation of China (Grant No. 41671149), and the Chinese Academy of Sciences Strategic Priority Research Program (Grant No. XDA19040101).

**Conflicts of Interest:** The authors declare no conflict of interest.

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
