# Peer review of "Spatially Explicit Reconstruction of Anthropogenic Grassland Cover Change in China from 1700 to 2000"

_land, doi:10.3390/land9080270_

Round 1

Reviewer 1 Report

This is a very competent manuscript. It is very well written and presented.  It has a logical flow and professional and informative tables and figures. There is a very extensive set of appropriate references.

The topic of historical grasslands in China is very specialized and of limited interest to the scientific community but does add to the similar regional studies.  It also provides a model for similar investigations.  The authors did collect and analyze an extensive amount of data.  Estimating historical conditions is often speculative but the authors made viable attempts to verify their results.

As in most manuscripts, there are some editorial suggestions for consideration by the authors, several of which follow:

  1. The use of personal pronouns (we, our) in scientific text varies by scientists and journals and this reviewer prefers they not be employed but that is really an editorial decision
  2. It would be helpful to define grasslands and the various types of grasslands in the text such as, meadow, and secondary. Grasslands are often viewed as a transition between desert shrub and savanna.
  3. Line 101, define TM, ETM.
  4. There are minor inconsistencies in the format of the references. The journal title in 5 could be abbreviated.  Article titles in 7 and 19 in lower case.   The use of et al. for authors in 17, 62 and 63 seems inconsistent.

In summary, this is a very competent manuscript and very suitable for Land.

Reviewer 2 Report

Conceptual and Methodology:

The manuscript presents an interesting approach for using secondary and historical data to develop a 300-year assessment of changes in grassland cover over eastern and western China. Figure 2, although is vague in some aspects, manifests the adopted approach for using logical operations over existing data for forest and cropland land covers.

In that aspect, the authors kept referring to “historical documents” as the source for this study as well as other studies as detailed in section 2.1. My main concern here is that the authors did not provide any information on what is meant by “historical documents” and what do these documents contain?

  • Are these records of ownership or taxation, or are they newspaper clips?
  • Are they descriptive or spatially explicit, detailed or summative?
  • What are the levels of details regarding land cover categorization, the temporal sampling, and spatial resolution?
  • What is the level of confidence and reliability of the information in these documents?

The reader is left with the impression that all data would be combined to provide one static view for the grassland cover. However, the reader will be shocked when seeing the temporal graphs in Figure 4 and reading in section 3.2 that the authors can identify 1950 as the year of change in the trend. Was the grassland reconstruction done at 50-year sampling? Does the data in the “historical documents” allow 50-year modelling/reconstruction at 10km resolution? These missing bits of details cast a doubt on the validity of the methodology and its results.

The other problem is that most of the references used in the adopted approach are actually prior studies by the some or all of the co-authors of this manuscript, specifically references number 39, 23, 40, and 26 (Table 1). Questions would be rightfully asked if it is proper or trustworthy when authors use the results of their own previous products as THE MAIN reference to support and validate their new results. I don’t think it is acceptable to expect the readers of this manuscript to go and read all those reference and judge for themselves.

A less critical issue is the lack of justification for specifying 90%, 85% and 90.22% to refer to the proportions of grassland area within the corresponding temperature, precipitation, altitude, and slope range (Table 2).

Comments on Discussion and conclusion:

Section 4.1 is just a historical corroboration of the results. However, according to what’s written in this section, the “historical documents” seems cannot provide any assertion or validity to the results at 50-year and 10km resolutions. Furthermore, lines 320-326 that discuss the changes after 1950 have no reference, but the reader has to take the authors narrative.

Section 4.2 regarding the comparisons with CLUD is scientifically weak. CLUD is used in the reconstruction of the grassland cover for the period after 1950, thus comparing the results with CLUD doesn’t make sense nor it provides validation. Actually, and considering that CLUD was established based on satellite remote sensing and supported/validated by field work, it must be considered as the ground truth. As such, the differences between CLUD and the authors reconstruction of the grassland should be considered failure by, or weakness in, the adopted approach.

Consequently, the statement on line 387, “our methods for reconstructing grassland cover are valid and provide a useful reference for the reconstruction of historical changes in other regions and in other types of land cover” ias not proved justifiable.

Typographical errors and suggestions:

Line 14: change “Progress” to “Studies”.

Lines 21-23: weakly structured/grammar sentence; revise.

Line 29: Change “Over the past thousands of years” to “Over the past” as there is no proof/support for this statement (although it is logical).

Line 32: “Long-term LULCC has therefore been considered as the main driver of global environmental change” I wonder if “the main driver” is a valid characterization. It is also true that environmental changes are a driver for LULCC.

Line 32: change “environmental change, and substantial” to “environmental change. Substantial”.

Line 97 Table 1: separation between rows is not clear in the 5th column (Reference).

Line 137: remove I, II, and III form Figure 1; they were never used in the text and they clutter the figure.

Line 248: re-arrange the sequence of lines in the legend to match their sequence on the graphs. Better yet, use different line symbols to present each graph line.

Line 383 Table 3: Wrong headings and separation between rows.

Reviewer 3 Report

Review for publication “Spatially Explicit Reconstruction of Anthropogenic 2 Grassland Cover Change in China from 1700 to 2000”

Publication entitled "The publication of the "Spatially Explicit Reconstruction of Anthropogenic 2 Grassland Cover Change in China from 1700 to 2000" is an interesting and valuable item in today's world, where the initiated changes in land cover and climate change have not come from nowhere and are mainly the result of human activities.

I appreciate that the authors of the publication also noted that "Long-term anthropogenic land use and land cover changes (LULCC) is regarded as an important component of past global change. The past 300 years have witnessed dramatic changes in LULCC in China, and this has resulted in the large-scale conversion of natural vegetation to agricultural landscapes"; this applies to the text of lines 11-14. Taking this reasoning into account, it can be concluded that any change in the environment should be considered as an adverse effect by current living people, excluding skillfully undertaken corrective actions and reconstruction of the land environment.

As far as literature is concerned, the publication contains many references, considered valuable and important by the scientific community. These are publications originating from China as well as from European countries and the Americas. This approach deserves my approval and has gained my approval. The vast majority of the relevant statements are supported by a large amount of scientific evidence.

As far as the cartographic material is concerned, however, I think it is of insufficient quality. Figure 1 (line 137) is illegible. Of course, it does not present the content relevant for publication, except for the model of terrain surface. The figure should be increased to the maximum size, the borders of the coastline and the border of the territory of China presented in a milder way. Small islands are illegible - they are almost completely covered by the contour. The WGS84 mapping grid markers are missing (or a different mapping layout - I leave it to the authors of the paper), and the north direction is missing. Similar descriptions should be found for the visible enlarged frame in the lower right-hand corner of the figure, according to the recommendations I myself receive from the MDPI Publisher.

Figure 2 is understandable to me, but it presents the methodology used by the authors in a general way and I believe that it cannot be reproduced in conditions other than here in this publication. Chapter 2.3 The methodology should be expanded to include a specific selected example from the borderline of the application of these two variants of reconstructing grassland cover in China. I would be very interested in a specific example of taking and fitting:

- historical cartographic materials to the model,

- Conceptualisation, discretisation and classification of historical descriptive materials, manuscripts, descriptions of nature, etc., and methods of their placement in the calculation model,

- data on geological analyses, based on dust testing methods,

- superimpose the results of analyses of current cartographic materials, e.g. from the land cover database,

- overlay and calculation of vegetation parameters resulting from modern LIDAR techniques.

A few such examples would allow for effective propagation of the method used by the authors in the world of science, which the current way of description does not guarantee, but even makes it difficult and possible to interpret differently, sometimes incorrectly.

Figure no. 3 is illegible, it is related to the comment I made in relation to figure no. 1. Why repeat the enlarged area in the frame when it is completely illegible? Only the coastline of the islands is visible. The WGS84 cartographic grid markers are missing (or a different mapping layout - I leave this to the authors of the work), and the north direction is missing.

In relation to figure 5 I have to make identical accusations. I propose to make a full-page drawing in a two-row system of map thumbnails, and in the lower right corner insert the legend in the 20% interval, not as it is now 10%. For such small maps it is not possible to read differences in the continuous scale from white to green. I ask the authors to consider a discrete leap classification every 20% - then any comparative interpretation would be possible. This is the scale used by the authors in the case of Figure 8.

In relation to figure no. 6 - I propose similar treatments as for figure no. 5. Color of differences are invisible, few pixels reach maximum values, so the whole is illegible. I propose to divide the analyzed parameter into 11 twenty percent intervals from -100% -- -5% -- +5% -- 100% with a separate middle eleventh class for which the differences are zero or fluctuate in a five percent range. All the figures also apply to my comments on the cartographic frame descriptions.

Figure 8 should be made more readable as in the suggestions for previous figures. I propose to change the colour scales used for visualisation of cartograms to the 20% range - they will be more legible and distinct - especially in case of figure 8C - showing the difference between the reconstruction and CLUDs. There is no need for these frames to enlarge an island area - if they should be, they should indicate the place of enlargement.

As far as the conclusions for publication are concerned, I believe that the discussion was presented in a reliable and comprehensive way, but still at a high level of generality. If the authors follow my recommendations for extending the description of the methodology, they will have additional opportunities to evaluate it in their proposals. The conclusions come from the content of the research, but do not present any inconsistencies, cases of exclusion of source data and a description of how to solve key cases. Did all calculations run so smoothly and according to general assumptions?

Reviewer 4 Report

The paper is interesting, however in my opinion neither original nor innovative, especially in comparison with the previous authors’ article (e.i. Yang, F.; He, F.N.; Li, S.C.; Li, M.J. Exploring spatiotemporal pattern of grassland cover in western China from 1661 to 1996. Int. J. Environ. Res. Public Health 2019, 16, 3160, doi:10.3390/ijerph16173160) . The authors are asked to provide information on the innovative and original aspect of their research, with particular emphasis on what they have already achieved.

The research goal, as well as the research question, are not given.

Fig 1 – nor readable.

There is a lack of comprehensive, including the study from other than China countries/regions.

No explanation why the study region is divided in two parts, description in 2.2. does not explain this issue unequivocally. Fig.1 shows three regions.

Fig.3- missing data- where there are? Pictures are too small

Lines: 388-389 – “Thus, our methods for reconstructing grassland cover are valid and provide a useful reference for the reconstruction of historical changes in other regions and in other types of land cover.” Lines 408-409 – “As shown in Figure 7d–f, the uncertainties in grassland area obtained by our method in eastern China are high.” How do you explain these two statements?

How do you deal with uncertainties in historical data, e.g. semantic in the grassland definition? How does it impact the study results?

The section Conclusion contains mainly summary. Must be improved.

In my, opinion the paper needs substantial improvement, especially regarding the originality (and a new aspects) of research (methods and results) and its scientific relevance? 

Round 2

Reviewer 2 Report

The Authors were successful in addressing all issues listed in my previous report.

However, reading the manuscript again after correction still raise some unease about certain aspects. Starting from the easier issues:

1- Table 1 should have a clear separation between rows. Reference [41] should not be on the same line as CLUDs.

2- The word “modern” (line 97, 175, 184, 186, 192, 195, 196, 223, 234, 353, and 369) gives a different connotation than what it was intended for. I suggest changing these incidents into “recent”, “current”, or “contemporary” as these words are more commonly used to refer to current/contemporary datasets in the literature.

3- Revise and improve the description of the steps in the methodology in Section 2.3.1 to be consistent with the terminology and flow used in Figure 2.  

4- One issues from my previous report that I wish the authors would address it more convincingly. That is, what are the levels of confidence and reliability of the information in the CHCD, CHFD, and GPNV? How is the uncertainty inherent in these datasets impact the results and the findings of the method?

I think addressing this point would be more relevant to the wider readership than the long qualitative/historical corroboration of results presented in Section 4.1. By the way, I still believe that this section should be shortened into a more concise version.

Reviewer 3 Report

The vast majority of my comments were taken into account in the second version of the publication. It is a pity that my main comment has not been included in the text, it could certainly have been applied and verified many times by other researchers or in other parts of the world. However, the explanation is clear and understandable, so I will not ask for further corrections. I conclude that this article is suitable for publication.

Author Response

Thank you for your valuable comments on this manuscript. Combined with other reviewers' opinions, the newly uploaded manuscript after detailed modification can be consulted.

Reviewer 4 Report

The authors introduced quite significant changes to the text and explained my doubts about the originality of the research. Therefore, I have nothing against publishing the article as it stands.

Author Response

(The authors gave the same response as above.)
